# Primary Treatment Results in Patients with Ovarian, Fallopian or Peritoneal Cancer—Results of a Clinical Cancer Registry Database Analysis in Germany

**DOI:** 10.3390/cancers14194638

**Published:** 2022-09-24

**Authors:** Robert Armbrust, Peter Ledwon, Anne Von Rüsten, Constanze Schneider, Jalid Sehouli

**Affiliations:** 1Department of Gynecology with Center for Oncological Surgery, Virchow Campus Clinic, Charité Medical University, 10117 Berlin, Germany; 2Gynecolocigal Tumor Center Berlin, 13353 Berlin, Germany; 3Clinical Cancer Registry for Brandenburg and Berlin, 03044 Cottbus, Germany

**Keywords:** ovarian cancer, primary surgery, adjuvant chemotherapy, lymphadenectomy

## Abstract

**Simple Summary:**

Diagnosis and treatment of ovarian cancers has changed significantly over the last years. However, the role of primary surgery and chemotherapy remain important parts of the multimodal treatment. Furthermore, real life data are often lacking but are very important for improving quality indicators and for hypothesis generation for future trials. The present work represents the first major analysis of federal cancer registry data of OC patients in Germany. Overall, 2771 primary OC cases were included. The results clearly elucidate quality measurements and treatment results and show good treatment outcomes in patients with primary OC compared to other internationally reported outcomes.

**Abstract:**

Background: The current therapy of ovarian cancer is based on the so-called “Three-Pillar-Model”, consisting of surgery, chemotherapy and maintenance therapy. This study represents the first major analysis of a federal cancer database of OC patients from the states Berlin/Brandenburg in Germany. The primary objective was to evaluate the prevailing established quality indicators surgical outcome, adjuvant chemotherapy and integrity of surgical staging in early stages. Methods: Data from the Clinical Cancer Registry for Brandenburg and Berlin of the years 2009–2019 were analyzed. Objectives were defined by a working group of selected physicians. Descriptive statistics were performed, as well as survival analysis. Results: A total of 2771 primary OC cases were included. Results regarding histological subtype met the suspected allocation with predominantly high-grade serous OC in advanced stage. The rate of complete surgical staging in FIGO stages I–IIA was 57%, and the rate of macroscopic complete resection in >FIGO III was 53%. Five-year survival rate varied from 79% (FIGO I) to 40% (FIGO III). Rate of adjuvant chemotherapy was above 50%. Conclusion: The results elucidate quality measurements and treatment results and show good treatment outcomes in patients with primary diagnosis. However, they also indicate deficits and can help to establish new quality indicators to further improve the treatment.

## 1. Introduction

The diagnosis and treatment of patients with primary and recurrent ovarian cancer (OC) has changed and improved significantly over the last few years. In particular, the introduction of maintenance therapy concepts renewed the therapy strategies in OC patients [1,2]. However, the establishment of innovative and new therapy algorithms and concepts come along with new challenges in patient care and counseling and affect both physicians and patients. In Germany, currently about 7500 women are primarily diagnosed with ovarian, fallopian or peritoneal cancer. These are the fifth most common cancers among women in Germany, with an incidence of 4.8% after breast, colorectal, and lung and endometrial carcinoma. Despite the low incidence compared to other malignancies, OC remains the disease with the highest mortality among gynecological cancers [3,4]. Almost two thirds of the patients are already in advanced-stage disease, FIGO IIIB–IV, when primarily diagnosed [5]. This is mainly caused by the heterogeneity and non-specify of symptoms such as abdominal pain, but also due to the lack of early detection. Another cause is the deficit of screening programs which aim to improve disease-specific prognosis parameters such as overall or progression-free survival (OS and PFS). Data for OS and PFS are mainly derived by clinical trials or national cancer databases. However, so-called “real life data” from cancer registries are very important, as they help to detect gaps generally in the diagnosis and treatment, but they are also very supportive for generating new trial hypotheses based on the results of clinical trials.

The present paper represents the first analysis ever conducted of data derived from a cancer registry database of the two large German states Berlin and Brandenburg. The initiative of this analysis is based on a joint working group consisting of clinical experts from the “Project group Ovarian Cancer Berlin/Brandenburg” and the governmental cancer registry. The cancer registry collects several clinical and patient individual characteristics of patients with ovarian, fallopian and peritoneal cancer. In fact, this first analysis concentrated on the following aspects of treatment: correct surgical staging procedures in early-stage OC until FIGO stage IIA, with a special focus on the role of systematic pelvic and paraaortic lymphadenectomy and omentectomy, the rate of macroscopic complete resection in advanced-stage disease, the role of neoadjuvant and adjuvant chemotherapy and the role of pathological results regarding the grading (high vs. low grade).

Those main aspects were based on the well-known and published results about the influence on survival-specific parameters such as macroscopic complete resection or the importance of correct staging in suspected early stages.

## 2. Materials and Methods

### 2.1. Data and Methods

#### 2.1.1. Data Basis and Design of the Analysis

The data from the Clinical Cancer Registry for Brandenburg and Berlin, which started its joint work on 1 July 2016, was evaluated. While the area-wide clinical cancer registration in Berlin started on 1 July 2016, the clinical cancer registration in Brandenburg has a tradition of more than twenty years based on a voluntary agreement with the statutory health insurance companies.

Two different endpoints were considered for the present evaluation: On the one hand, a brief overview of ovarian cancer diagnoses (histopathological subgroups and FIGO stages) and their treatment (surgical procedures and systemic treatment) is shown by using data from the federal states of Brandenburg and Berlin and the diagnosis years 2016–2018. On the other hand, the 5-year overall survival according to FIGO stage and histopathological subgroup was analyzed. Since the evaluation of overall survival requires a longer follow-up period, only the data from the federal state of Brandenburg could be used. The diagnosis years 2009–2015 were considered for the survival analysis.

There is a special feature when recording the death data. For this purpose, a comparison was made with the death data of the common cancer registry of the federal states of Berlin, Brandenburg, Mecklenburg-Western Pomerania, Saxony-Anhalt and the Free States of Saxony and Thuringia (GKR). The GKR records all death data of the mentioned federal states by comparison with the death data of the registration offices as well as with the death certificates of the health authorities. Because of the data comparison with the GKR, the Clinical Cancer Registry also receives death data that are not directly reported to it, which ensures that all death data are fully recorded. At the time of the evaluation shown here, the death certificates had been completely processed by the GKR by the end of 2015, and these death data were accordingly transmitted to the Clinical Cancer Registry for Brandenburg and Berlin.

The analyses shown here are based on the data status from 12 August 2020 for the diagnosis, chemotherapy and death data and the data status from 3 October 2020 for the surgery data in the Clinical Cancer Registry for Brandenburg and Berlin.

#### 2.1.2. Inclusion and Exclusion Criteria

The selection of patients with ovarian, peritoneal and fallopian tube cancer was based on the following diagnosis codes: ICD-10: C56, C57.0, C48.1-.2 and ICD-O: C57.9. Reported diagnoses with the histology group sarcomas or germ line stromal tumors were excluded. Furthermore, in situ carcinomas or borderline malignant forms of ovarian, peritoneal and fallopian tube cancer were not taken into account for the present analyses.

For all questions related to therapy, the federal state of tumor treatment is considered, for the other questions (i.e., FIGO stage distribution and overall survival) the state of the place of residence (at the time of diagnosis) is taken into account. Unless otherwise specified, the place of treatment is defined as the federal state in which the first tumor-resecting surgery for ovarian, fallopian tube or peritoneal cancer was conducted. The tumor-resecting surgery was defined using the following OPS codes: 5-652, 5-653, 5-661, 5-682, 5-683.1/.2/.6/.7, 5-687, 5-651.8/. 9/.a, 5-665.4, 5-543.0/.1/.2/.4.

It also has to be highlighted that the number of cases varies in the analysis according to the specific inclusion criteria regarding the selected questions for this analysis. As a result, the total number of included patients was 1272 overall and consequently reduced for identified cases with “macroscopic complete resection, complete surgical staging in early stages, neoadjuvant treatment and adjuvant chemotherapy”.

#### 2.1.3. Specifications for the Classification of the Histopathological Grading

All diagnostic findings with grading G1 were classified as low grade, and those with grading G3 or G4 were classified as high grade.

In the case of G2, the assignment to the grading was based on the cell type. All serous or clear cell types were assigned to high grade in G2 and non-serous (mucinous, endometrioid) cell types in G2 were assigned to low grade. All unspecific or not clearly assignable histologies (e.g., adenocarcinoma NOS, carcinoma NOS) were excluded from the analyses in which the histopathological grading is relevant. For all other analyses that do not take into account the histopathological low- and high-grade subtype, unspecific or not clearly assignable histologies could be taken into account.

#### 2.1.4. Determination of Selected Therapies and Macroscopically Complete Resection

The individual operative interventions were defined using OPS codes. The systematic lymphadenectomy was defined as follows: OPS code 5-402, 5-406, 5-404, 5-407, 5-685.1/.2/.3 and 5-685.41/.42/.43 as well as 5-686.1/.2/.3. In addition, the indication of >2 examined lymph nodes, even without a corresponding OPS code, was rated as a systematic lymphadenectomy. Omentectomy was defined using the OPS code 5-543.2. The number of requested therapy procedures was counted as the number or proportion of tumor cases with the corresponding procedure.

Macroscopically complete resection was defined as the global R0 or R1 category. If no global R category was given in the notification of surgery, but a local R0 category (without metastases) or full remission was reported at the same time, it was also assumed that the tumor was macroscopically complete resected. The information on full remission was obtained from the reported data about course of the tumor.

With regard to chemotherapy, all patients with a documented start of internal therapy were considered as cases with chemotherapy. In addition, the position of chemotherapy in relation to surgery was determined by comparing the date of beginning of internal therapy with the date of the first tumor resection.

#### 2.1.5. Statistical Analyses

Analyses were performed separated by FIGO stage or histopathological grading. Absolute and relative frequencies were calculated. Overall survival (OS) was calculated as the difference between the date of diagnosis and the date of death (from any cause) or the cut-off date 31 December 2015. Survival time data were analyzed using Kaplan–Meier curves, and 60-month probabilities were presented according to FIGO stage and histopathological grading. Survival curves were compared with the log-rank test. The significance level was fixed at 5% and the analyses were performed in SPSS, version 24, IBM.3. In addition, RStudio, version 4.1.0, (package *survminer*) was used to produce the survival plots including 95% confidence intervals and number at risk.

## 3. Results

In the examined diagnostic period from 2016 to 2018, 1500 cases with a primary ovarian, fallopian tube or peritoneal cancer residing in the federal state of Brandenburg and Berlin were documented, and for the period of 2009 to 2015 a total number of 1303 were documented (excluding sarcomas, germ line stromal tumors or in situ and borderline malignant carcinomas). For further detailed information about selection criteria and the selection process, see Figure 1.

### 3.1. Results of FIGO Stages, Histology and Grading

Due to further restricting to serous, non-serous and clear cell carcinomas, the number of cases was reduced to 972. The majority of these patients were FIGO stage III (47%) for high-grade carcinomas and for low-grade carcinomas; on the other hand, FIGO stage I predominated with 55% (Figure 2 and Table 1). The most common histology was serous.

### 3.2. Surgery for Tumor Staging and Rates of Macroscopically Complete Resection

The analysis of surgical interventions comprised data from 919 patients (reduced number of cases according to inclusion criteria “tumor staging” and “macroscopic complete resection”) of FIGO stage I–IV with the years of diagnosis 2016–2018 and treatment location in the federal state of Berlin or Brandenburg.

The percentage of systematic lymphadenectomy performed in relation to the operated patients in the early FIGO stages I to IIA was 57% in Brandenburg and 60% in Berlin. There were no clear differences between early and advanced stages or between Berlin and Brandenburg (Figure 3). The frequency of an omentectomy in the early FIGO stages was 68% in Brandenburg and 75% in Berlin (Figure 4).

An overview of the rate of macroscopically complete resection for FIGO stages I to III is shown in Figure 5 in a comparison of the two federal states as the treatment location. As expected, there are clear stage-specific differences, with correspondingly higher rates of macroscopically complete resection in the earlier stages. Concerning the frequent stage FIGO III, the rate of macroscopically complete resection was 53% in Brandenburg and 45% in Berlin. However, the high proportion of cases with no information on the resection status should be noted here.

### 3.3. Systemic Therapy

To address the question of the frequency of systemic therapy, separated by FIGO stage, data from both federal states from a total of 1259 patients and diagnosis years 2016–2018 were evaluated. Unlike the analyses related to surgical procedures (Section 3.2), the place of treatment was defined as the federal state of first tumor resection or, in cases without surgery, the federal state of chemotherapy, radiation or diagnosis. Chemotherapy was documented in 54% of all patients with treatment location in Brandenburg or Berlin across all stages. In relation to all patients, the proportion of reported and documented neoadjuvant chemotherapies was 6% in total and the proportion of adjuvant chemotherapy across all stages was 41%. Within the group of patients who received adjuvant chemotherapy, 81% had a documented start of treatment within 8 weeks after the first tumor resection. It was shown that in advanced stages the proportion of patients receiving chemotherapy within 8 weeks after the primary tumor resection was higher than in the early stages.

### 3.4. Results of Survival Analysis

Both FIGO stage and the grading are relevant prognostic factors for overall survival (Figure 6). In FIGO stage III the absolute 5-year survival rate is 40% compared to 28% in FIGO stage IV, in the early FIGO stages I and II the absolute 5-year survival was 79% and 67%, respectively. A comparison between high- and low-grade carcinomas also showed a clear difference of 42% absolute 5-year survival rate for high-grade carcinomas in contrast to 68% for low-grade carcinomas. Survival data were only available from the Brandenburg Register for the years 2009–2015.

## 4. Discussion

The present analysis represents the first major data evaluation from the federal cancer registry databank of the states Berlin and Brandenburg regarding several clinical aspects in the treatment of patients with primary ovarian cancer. Despite single analysis from other federal states or analysis due to quality certification processes, this is a result of a joint working group consisting of clinical and scientific experts. A similar comprehensive assessment from over 600 patients with ovarian cancer was conducted in the federal state of Bavaria. However, the results are not available as a full-text publication yet, and are currently only presented as a poster on national congress [6].

Regardless of several new findings in the systemic treatment of ovarian cancer, primary upfront debulking surgery still plays a major role in the primary as well as in the recurrent treatment of OC. Surgery is therefore an important part of the so-called “Three-pillar-strategy” consisting of surgery, systemic chemotherapy and maintenance treatment. In particular, macroscopic complete resection of all visible intra-abdominal tumors has a significant influence on PFS and OS both in primary and recurrent disease.

However, there are still questions to answer. What is the optimal timing of debulking surgery? What are the best instruments to preoperatively evaluate patients feasible for macroscopic complete resection? How can patients be optimally assessed and possibly optimized regarding the risk for surgical-associated morbidity and mortality? All these questions should be considered carefully within the treatment and have significant influence on survival outcome measurements such as PFS and OS. The present analysis could demonstrate the high quality in standard of care of OC patients among the two federal states Berlin and Brandenburg regarding nationally and internationally published and well-established quality measurements [7,8]. There is also recent evidence that cancer mortality rates are decreasing ion Europe [8]. The stage-dependent five-year survival rate in the common FIGO stage III is 40%, compared to previously published data [9]. In addition, the rate of macroscopic complete resection within primary upfront debulking surgery in advanced-stage disease is on a high level [10]. However, our data also show weakness in the correct documentation of macroscopic tumor residuals, and this is therefore a significant limitation of the study. According to international recommendations, the surgeon should make a clear statement on the postoperative visible tumor residuals. This should be reported with the following terms: macroscopic complete resection, macroscopically visible residuals <10 mm or macroscopically visible residuals >10 mm. The cancer registry databank of Berlin/Brandenburg captures this via TNM Classification with R Status (R0, R1, R2), which should be obsolete in OC. This is a major limitation of the current database and will be changed prospectively upon the results of this analysis for future data capture.

Regarding the two major topics of this cancer database evaluation, the completeness of correct surgical staging in early stages and rate of adjuvant therapy, the results are inhomogeneous: the stage-dependent rate of systematic paraaortic and pelvic lymphadenectomy varied in both states between 52% and 61% and is therefore below the expected level. This is especially conflicting under the view of the importance of correct indication and performance of systemic lymphadenectomy and the significance on possible up-staging: previously published data could demonstrate that patients with an “incomplete” staging in early-stage OC have a significantly worse PFS and OS (5-year PFS 79% vs. 61%, 5-year OS 89% vs. 71%) [11,12,13,14,15,16]. However, it is not possible from our data to distinguish between a possible reporting problem or issue of the centers and the actual rate of complete staging. A possible argument for reporting issues might therefore be the high and unsuspected variation in the numbers. Similar varying and suboptimal results were observed regarding the frequency of an adjuvant systemic treatment in advanced stages. However, these numbers should also be interpreted with caution, as adjuvant treatment in Germany is often not performed by the center or hospital performing the surgery and is therefore often located in an outpatient setting. As a result, the authors therefore suggest that the low numbers are mainly explained by a lack in reporting. The results also show that the concept of neoadjuvant treatment in primary OC only plays a minor role in Berlin and Brandenburg.

## 5. Conclusions

These “real world” results clearly elucidate quality measurements and treatment results. In addition, the good results regarding survival outcomes and treatment characteristics of the analysis also could clearly reveal weaknesses in the reporting process and the consistency of such a database. The current database should therefore be renewed and changed, especially regarding the introduction of new treatment concepts such as the above-mentioned maintenance treatment, but also aspects of cancer aftercare (e.g., problems of long-term cancer survivors). Another reason for the need of continuously updating the national cancer registry database is the rapid growth of newly published treatment guidelines and their ongoing update process. For example, in 2020 the ESGO published an updated version of quality indicators for advanced ovarian cancer surgery. Those results should be implemented in cancer database registrations and could therefore help to better picture the general quality in diagnosis and treatment of OC patients [7].

## Figures and Tables

**Figure 1 cancers-14-04638-f001:**
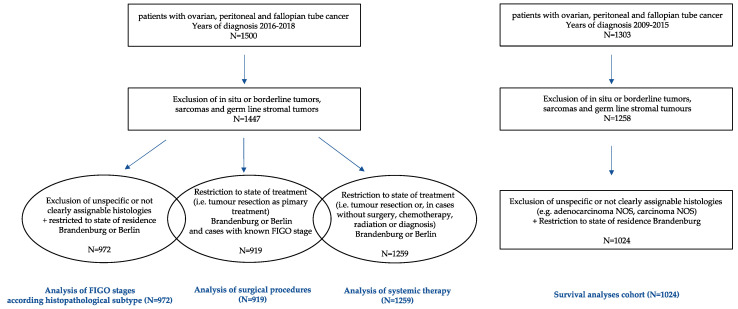
Flow Chart of selection criteria and process, detailed information on ICD and OPS Codes are provided as Appendix A at the end.

**Figure 2 cancers-14-04638-f002:**
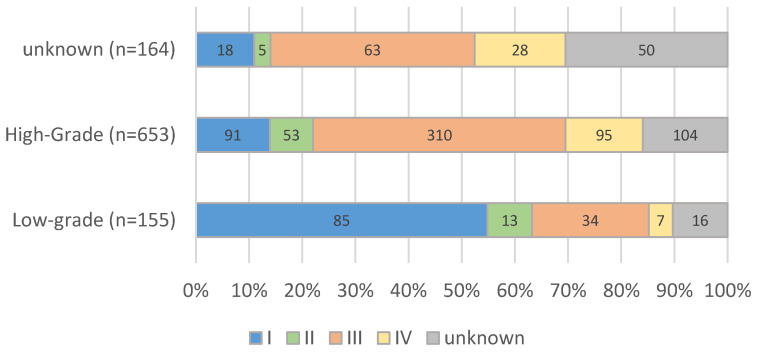
FIGO stages according to histopathological subtype, years of diagnosis 2016–2018 (*n* = 972, years 2016–2018).

**Figure 3 cancers-14-04638-f003:**
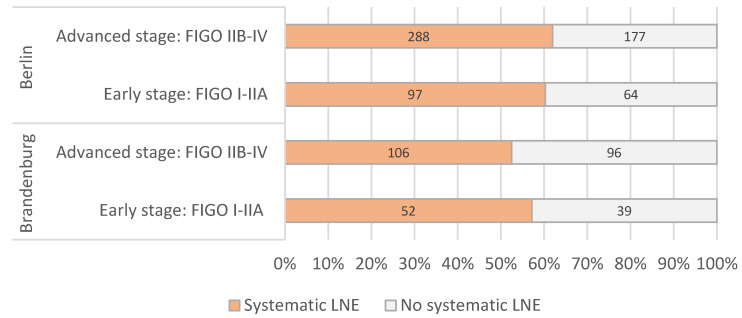
Proportion of patients with primary ovarian cancer with and without systematic pelvic and paraaortic lymphadenectomy according to FIGO stage (*n* = 919, years 2016–2018).

**Figure 4 cancers-14-04638-f004:**
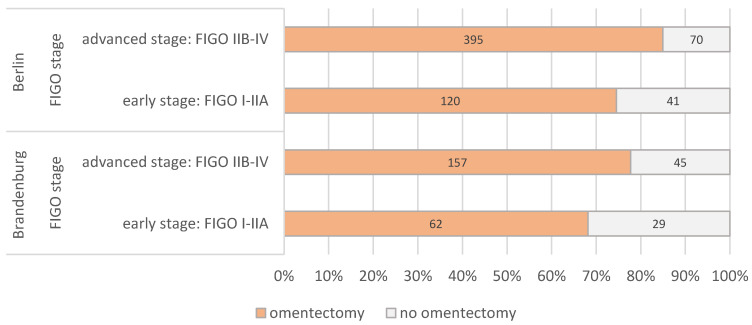
Proportion of patients with primary ovarian cancer with and without omentectomy according to FIGO stage (*n* = 919, years 2016–2018).

**Figure 5 cancers-14-04638-f005:**
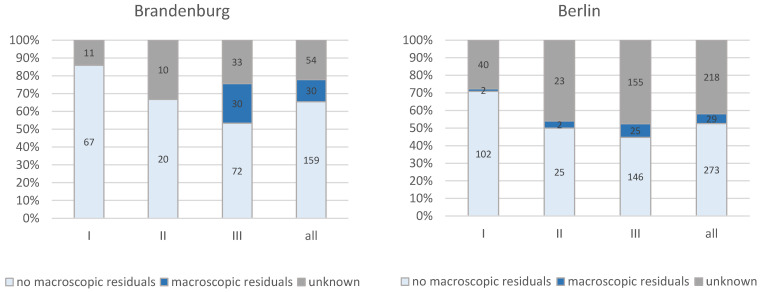
Percentages of macroscopic complete resection according to FIGO stages and federal states Berlin vs. Brandenburg (*n* = 763; FIGO stage I: *n* = 222, FIGO stage II: *n* = 80, FIGO stage III: *n* = 461).

**Figure 6 cancers-14-04638-f006:**
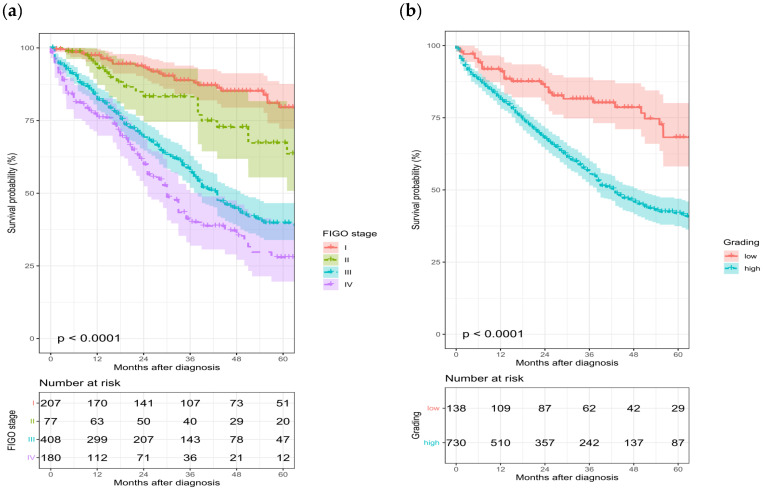
The 5-year survival according to FIGO stage ((**a**), *n* = 872) or grading ((**b**), *n* = 868), years of diagnosis 2009–2015.

**Table 1 cancers-14-04638-t001:** General cohort characteristics.

	Years of Diagnosis 2016–2018, Patients from Brandenburg and Berlin (*n* = 972)		Year of Diagnosis 2009–2015, Patients from Brandenburg (*n* = 1024) ^1^	
Median age (years)	66.3		67.7	
*State of residence*	*n*	%	*n*	%
Brandenburg	478	49.2	1024	100.0
Berlin	494	50.8	NA	
*FIGO stage*				
I	194	20.0	207	20.2
II	71	7.3	77	7.5
III	407	41.9	408	39.8
IV	130	13.4	180	17.6
not specified	170	17.5	152	14.8
*Grading*				
low	155	15.9	138	13.5
high	653	67.2	730	71.3
unknown/not specified	164	16.9	156	15.2
*Histology group* ^2^				
Serous	798	82.1	809	79
Non-serous (endometrioid, mucinous)	140	14.4	179	17.5
Clear cell	34	3.5	36	3.5

^1^ This cohort was used for the survival analysis only and is shown for comparison. ^2^ Germ cell tumors and sarcomas are excluded from all evaluations. In addition, all forms that cannot be clearly assigned to the serous, non-serous or clear cell subtype are excluded in the evaluations, in which the histopathological grading is relevant (i.e., survival or FIGO stage distribution according to histopathological subtype).

## Data Availability

The data can be shared up on request.

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
