# Peer review of "Primary Treatment Results in Patients with Ovarian, Fallopian or Peritoneal Cancer—Results of a Clinical Cancer Registry Database Analysis in Germany"

_cancers, 2022, doi:10.3390/cancers14194638_

Round 1
Reviewer 1 Report (Previous Reviewer 3)
Dear authors, thank you for the changes. Overall, this could be a nice piece of information but methodological improvements needs to be done from my point of view. I think you cannot describe survival in a cohort of patients without describing FIGO, age, histology... in the same cohort (as far as I understood descriptions belong to an independent cohort from 2016 to 2018). I would advice you to describe both cohorts separately, only the first with survival outcomes, and then compare changes in FIGO, surgical results...along time between the two cohorts, driving conclusions on potential changes in the stadification process and surgical imrpovemtns along the study period. Additionally I don't understand why N is 1272 in the Section 3.2, instead of 972. Some other comments: I think grammar and English still needs revision, and abstract is very brief (could be enhanced with more information). I don't understand Line 147-148, please clarify. All my best,
Author Response
Reviewer 1:
Dear authors, thank you for the changes. Overall, this could be a nice piece of information but methodological improvements needs to be done from my point of view. I think you cannot describe survival in a cohort of patients without describing FIGO, age, histology... in the same cohort (as far as I understood descriptions belong to an independent cohort from 2016 to 2018). I would advice you to describe both cohorts separately, only the first with survival outcomes, and then compare changes in FIGO, surgical results...along time between the two cohorts, driving conclusions on potential changes in the stadification process and surgical imrpovemtns along the study period.
We now added a Table 1 to the Results section describing general characteristics (age, FIGO stage, histopathological subtype) of the two samples: patients from Brandenburg and Berlin, years of diagnosis 2016-2018 vs. patients from Brandenburg, years of diagnosis 2009-2015.
Of note, we have no patients residing in Berlin before 2016. Furthermore, the completeness of the data is varying over time, which limits the comparison of treatment quality (especially for macroscopic complete resection vs. macroscopic residuals). Due to this limitation, we decided to do not address the comparison of surgical treatment and outcome at different time periods in the manuscript.
Additionally I don't understand why N is 1272 in the Section 3.2, instead of 972.
First, the differing sample sizes are a result of differing selection criteria, which were defined according to the specific research question. Second, the originally described sample size of 1259 instead of 1272 is correct for section 3.2. (this is modified in the manuscript now).
N=972, for instance, is the sample size of the The FIGO stage distribution. Here the state of the place of residence (at the time of diagnosis) is taken into account.
For questions related to therapy, especially surgical procedures, the federal state of tumour treatment is considered (see section 3.1).
In contrast, N=1259 is the sample of the evaluation of the frequency of systemic therapy (section 3.2). Here the place of treatment was considered, which was defined as the federal state of first tumour resection or, in cases without surgery, the federal state of chemotherapy, radiation or diagnosis.
We now also added a sentence in section 3.2 (line 220-222) to clarify the differing selection criteria.
Some other comments: I think grammar and English still needs revision, and abstract is very brief (could be enhanced with more information).
We checked and corrected again several errors. Furthermore the abstract is limited to 200 words, that why we kept it short.
I don't understand Line 147-148, please clarify. All my best,
This lines refers to the following sentence “The information on residuals was obtained from the reported data about course of the tumour.” ïƒ This sentence was now modified to clarify that we refer to the information whether the tumour was completely resected/complete remission: “The information on full remission was obtained from the reported data about course of the tumour.”
Reviewer 2 Report (New Reviewer)
The authors present registry data from ovarian/fallopian/primary peritoneal cancer patients treated in the German states Berlin/Brandenburg. The manuscript is well written and documented, concise, and informative.
My only suggestion is to add numbers at risk under the X axis, and shaded areas representing 95% confidence intervals to the curves, to the KM plots.
Author Response
Reviewer 2:
The authors present registry data from ovarian/fallopian/primary peritoneal cancer patients treated in the German states Berlin/Brandenburg. The manuscript is well written and documented, concise, and informative.
My only suggestion is to add numbers at risk under the X axis, and shaded areas representing 95% confidence intervals to the curves, to the KM plots.
We thank the reviewer for that comment. The KM plots are now produced using RStudio instead of SPSS, which allows for additionally representing the 95% confidence intervals to the curves as well the number at risk under the X axis.
Round 2
Reviewer 1 Report (Previous Reviewer 3)
Dear authors,
Some things are now clearer to me. Thank you for answering item by item to my requests. I have some further major and minor questions that i am sure you will easily address.
Minor: typo error line 38 (primary surgery )
Major:
1. Can you add Histology in Table 1?
2. I think that grading and FIGO analysis should be under subtitle 3.1, and then surgical analysis under subtitle 3.2...and so on. In this section, could you clarify which further restrictions criteria were used to justify that you analyse 972 patients out of 1272, instead of 1272 which were already selected by histologic criteria mentioned in methods?
3. Because prognosis is based in a different population, I 'd suggest to describe survival analysis in another section (3.4, at the end). In this section, a sentence on the total number of patients analysed (1024) , remembering the period and geographical area, and also the inclusion criteria required, would be necessary at the beginning of the paragraph (similarly as you have already done in line 156 for the FIGO and grading analysis).
I think these changes are necessary to clarify methods to non-German readers, which we are not used to assurance and geographical issues of Germany. Highlighting that survival analysis is done in a different population with longer follow up (and thus with very strong survival data) can be seen as a strong point of the research which states the initial point where we stood some years ago. Also, histopathological distribution are very similar among the two population, which highlights the importance of surgical and systemic therapies, and it will be very interesting to see survival outcomes of the more recent population in the next 5 years.
4. Is data on performed laparoscopies with staging purposes available? if you have it, it would be interesting.
thank you a lot, best wishes,
Author Response
Reviewer 1:
Thank you again very much for commenting on our paper. We want to address each single comment.
Minor: typo error line 38 (primary surgery )
Answer: we corrected this typo.
Major:
- Can you add Histology in Table 1?
We added the Histology in Table 1. Thank you for this useful comment
- I think that grading and FIGO analysis should be under subtitle 3.1, and then surgical analysis under subtitle 3.2...and so on. In this section, could you clarify which further restrictions criteria were used to justify that you analyse 972 patients out of 1272, instead of 1272 which were already selected by histologic criteria mentioned in methods?
We corrected the order of the subtitles according to your suggestions.
- Because prognosis is based in a different population, I 'd suggest to describe survival analysis in another section (3.4, at the end). In this section, a sentence on the total number of patients analysed (1024) , remembering the period and geographical area, and also the inclusion criteria required, would be necessary at the beginning of the paragraph (similarly as you have already done in line 156 for the FIGO and grading analysis).
We also put in a new section with results of the survival analysis unter subtitle 3.4. Also we again explained the differences regarding the differing number of cases in the methods section according to you suggestions.
I think these changes are necessary to clarify methods to non-German readers, which we are not used to assurance and geographical issues of Germany. Highlighting that survival analysis is done in a different population with longer follow up (and thus with very strong survival data) can be seen as a strong point of the research which states the initial point where we stood some years ago. Also, histopathological distribution are very similar among the two population, which highlights the importance of surgical and systemic therapies, and it will be very interesting to see survival outcomes of the more recent population in the next 5 years.
Thank you very much for this comment, we agree and will also put work in that registry for future analysis answering those questions.
- Is data on performed laparoscopies with staging purposes available? if you have it, it would be interesting.
This is a very good point but unfortunately the cancer registry does not obtain data regarding laparoscopies.
Round 3
Reviewer 1 Report (Previous Reviewer 3)
Dear authors,
thanks for addressing most of the issues. Now much clearer. However, I think the following steps are needed:
1) a Flowchart of included /excluded patients in total and for each objective not related to survival, in which you state what means each code (ICD, surgical codification..). A list of codes with their title is needed to understand which patients you included or excluded at each step.
2) In the last section , regarding survival, please state that this population is from the Brandenburg register from 2009-2014, as you did at the beginning of the Results with the descriptive cohort (already required in prior report).
I believe these are the last comments from me. Tthank you for your efforts
Author Response
Dear Reviewer,
thank you again for your useful comments.
To Point 1:
We provided now a flowchart with the exact number of in- or excluded patients as a new figure. However, we don´t believe that it is useful to integrate all ICD 10 Codes and OPS Codes, mainly because the Figure could become confusing and incomprehensable. Furthermore, non german reader might not be familiar with surgical OPS Codes and the whole coding system in Germany. As a compromise we provided each single ICD 10 and OPS Code with meaning or explanation as supplemental material.
To Point 2:
We included a sentence in section 3.4 to clarify the issue regarding the available survival data.
We really hope that everything is now clearer to you and we are looking forward to publication. Thank you again very much for your comments and the effort!
This manuscript is a resubmission of an earlier submission. The following is a list of the peer review reports and author responses from that submission.
Round 1
Reviewer 1 Report
The present work represents the first major analysis of federal cancer registry data of OC patients in Germany. The results describes quality measurements and treatment results, showing good treatment outcomes in patients with primary OC. Although well written and presented, the study does not add any novelty to the scientific literature, being only a 'cross section' of the German clinical management of OC. Besides the good results regarding survival outcomes, the analysis clearly reveal weaknesses in the reporting process. In particular a very little space has been addressed to the chemotherapeutic approach, without citing type of chemorapeutic regimens and maintenance treatments, as well as aspects of cancer aftercare
Reviewer 2 Report
Thank you for asking me to review the manuscript entitled: “Primary Treatment Results in Patients with Ovarian, Fallopian or Peritoneal Cancer - Results of a Clinical Cancer Registry Database Analysis in Germany” by Armbrust et al.
This is a retrospective analysis of registry data from Berlin and Brandenburg, Germany aiming to assess the rate of surgical staging in early ovarian cancer, complete cytoreduction in advanced ovarian cancer and survival.
Overall, 2771 cases with primary ovarian cancer were retrieved from the registry and analyzed: 57% of all women with Stage I-IIA had complete surgical staging and 53% with FIGO stage III or higher were optimally cytoreduced.
This is an interesting study, but has several limitations:
Residual tumor burden per se is not an unequivocal objective assessment. In addition, the authors have only used surrogate parameters to define cytoreduction as this has not been entered into the database (e.g. the authors have used R0, R1 and R2 from the reports to assume residual disease).
Data were used from 2009-2019: New treatment modalities (PARP-inhibitors) have been incorporated and these have not been documented in the registry precluding any statistical approaches accounting for these.
Line 50: Should be rephrased, e.g.: Almost two thirds of the patients are diagnosed in advanced disease stages.
Line 79: The fact that Berlin data are only available after 2016 may introduce a bias as the level of complete cytoreduction appears to differ (Fig 4). This may affect long-term survival.
Line 164: “Surgery for tumor staging and rates of macroscopically complete resection” I assume this should be a headline?
All Figures: Figure legends need to be larger, can hardly be read.
Line 217: incomplete legends your reference says“Quelle DKG und ESGO Guideline, noch ergänzen“
Line 270: Can you confirm? informed consent was obtained from all participants that were entered in the federal registry?
Line 273: Please revise acknowledgement section - this seems to be the template.
Overall the article would benefit from an English review.
Reviewer 3 Report
Thank you very much for this effort, which is huge for us clinicians. Now that ovarian cancer mortality seem to decrease in EU countries (Dalmartello et al, Annals of Oncol 2022), this is a timeline manuscript to review wide-European regions' results, and to underline the importance of proper databases with clinical attributes to evaluate quality of our Medical practice. Using proper endpoints and survival surrogates, these registries could enable us to improve survival outcomes in the future. Please find my suggestions below, which will require a major revision, but I think it can be worthwhile . I would also include Dalmartello's paper as a reference that you can comment.
Thank you very much for this effort, which is huge for us clinicians. Now that ovarian cancer mortality seem de decrease in EU countries (Dalmartello Ann Onco 2022), this is a timeline manuscrpt to review wide-European regions results, and to underline the importance of proper databases with clinical attributes to evaluate quality of our Medical practice. Using proper endpoints and survival surrogates, these registries could enable us to improve survival outcomes in the future. Please find attached my suggestions, which will require a major revision , but I think it can be worthwhile . I would also include Dalmartello paper as a reference that you can comment.
In the abstract
Line 27-28: don’t understand what autors mean
Line 33: first sentence of conclusions incomplete
I would not say that OS aroung 40% in Advanced stages are goo results, maybe yes in comparison other country -based historical registries (i.e. SEER) but patients reading this article would not agree that 40% 5 years is a good results!
Main manuscrit:
Methods
Line 40-41: I ‘d simplify the second sentence: I’d suggest: “ Every part of the multimodal therapies used have been reevaluated , both surgical techniques and systemic treatments.”
Line 50: “Two ThirdS”
Line 53 and 54: I’d say “Another CAUSE is the deficit of screening programs which AIM TO improve disease specific prognosis parameterS like overall or progression free survival (OS and PFS).”
Line 64-66: This two sentences seem to contain the same message, can you choose one of them? “ The Cancer Registry collects several clinical and patient individual characteris-64 tics of patients with Ovarian, Fallopian and Peritoneal Cancer. The Database included 65 several clinical attributes as well as patient individual features (of patients with Ovarian, 66 Fallopian and Peritoneal Cancer). “
Line 95-97: not sure of what autors mean: data is updated until two diferent dates for systemic treatments and surgical procedures? Why? Please, clarify or simplificate. Later on, you specify that cut off data for survival is 2015.
2.1.1. Section: if I understood well, in case of data not available in registries, such as RO/R1 sometimes, medical records were consulted. I’d suggest to add this piece of information in 2.1.1 section, if correct.
I would add a first section in Methods entitled “Design”. I think this would help to understand the diferent endpoints: on the one hand “survival outcomes of ovarain cancer” – data from 2009 to 2015 Brandenburg; on the other hand “snapshot of ovarian cancer diagnsoses and how this is treated” (pathological, stages, surgical procedures and systemic treatmetns ) – data from 2016 to 2019 Brandenburg + Berlin. Am I right?
Furthermore, if possible I ‘d suggest to compare OS from two periods: 2009-2012 and 2013 -2015. Also I’d suggest to repeat the snapshot from at least the first period (2009-2012). This analysis could reveal diferent changes or trends in real life practive over the last decade.
Results
Line 162: this is a Minor Title, please put it in itàlics or bold...
Figures 3 and 4: please replace Stadium by Stage
Section 3.1: why now you use 1272 cases?
Line 217: German words “noch ergänzen”??